# Perfect density models cannot guarantee anomaly detection

## Abstract

Thanks to the tractability of their likelihood, some deep generative models show promise for seemingly straightforward but important applications like anomaly detection, uncertainty estimation, and active learning. However, the likelihood values empirically attributed to anomalies conflict with the expectations these proposed applications suggest. In this paper, we take a closer look at the behavior of distribution densities and show that these quantities carry less meaningful information than previously thought, beyond estimation issues or the curse of dimensionality. We conclude that the use of these likelihoods for out-of-distribution detection relies on strong and implicit hypotheses, and highlight the necessity of explicitly formulating these assumptions for reliable anomaly detection.

## 1 Introduction

Several machine learning methods aim at extrapolating a behavior observed on training data in order to produce predictions on new observations. But every so often, such extrapolation can result in wrong outputs, especially on points that we would consider infrequent with respect to the training distribution. Faced with unusual situations, whether adversarial (Szegedy et al., 2013; Carlini & Wagner, 2017) or just rare (Hendrycks & Dietterich, 2019), a desirable behavior from a machine learning system would be to flag these *outliers* so that the user can assess if the result is reliable and gather more information if need be (Zhao & Tresp, 2019; Fu et al., 2017). This can be critical for applications like medical decision making (Lee et al., 2018) or autonomous vehicle navigation (Filos et al., 2020), where such outliers are ubiquitous.

What are the situations that are deemed unusual? Defining these *anomalies* (Hodge & Austin, 2004; Pimentel et al., 2014) manually can be laborious if not impossible, and so generally applicable, automated methods are preferable. In that regard, the framework of *probabilistic reasoning* has been an appealing formalism because a natural candidate for outliers are situations that are *improbable* or *out-of-distribution*. Since the true *probability distribution density* $p_X^*$ of the data is often not provided, one would instead use an estimator, $p_X^{(\theta)}$, from this data to assess the regularity of a point.

Density estimation has been a particularly challenging task on high-dimensional problems. However, recent advances in *deep probabilistic models*, including variational auto-encoders (Kingma & Welling, 2014; Rezende et al., 2014; Vahdat & Kautz, 2020), deep autoregressive models (Uria et al., 2014; van den Oord et al., 2016b;a), and flow-based generative models (Dinh et al., 2014; 2016; Kingma & Dhariwal, 2018), have shown promise for density estimation, which has the potential to enable accurate *density-based methods* (Bishop, 1994) for anomaly detection.

Yet, several works have observed that a significant gap persists between the potential of density-based anomaly detection and empirical results. For instance, Choi et al. (2018), Nalisnick et al. (2018), and Hendrycks et al. (2018) noticed that generative models trained on a benchmark dataset (e.g., CIFAR-10, Krizhevsky et al., 2009) and tested on another (e.g., SVHN, Netzer et al., 2011) are not able to identify the latter as out-of-distribution with current methods. Different hypotheses have been formulated to explain that discrepancy, ranging from the *curse of dimensionality* (Nalisnick et al., 2019) to a significant mismatch between $p_X^{(\theta)}$ and $p_X^*$ (Choi et al., 2018; Fetaya et al., 2020; Kirichenko et al., 2020; Zhang et al., 2020).

In this work, we propose a new perspective on this discrepancy and challenge the expectation that density estimation should enable anomaly detection. We show that the aforementioned discrepancy

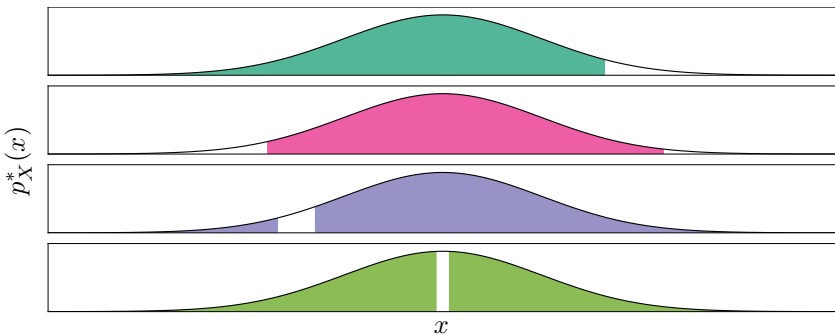

Figure 1: There is an infinite number of ways to partition a distribution in two subsets, $\mathcal{X}_{in}$ and $\mathcal{X}_{out}$ such that $P_X^*(\mathcal{X}_{in}) = 0.95$. Here, we show several choices for a standard Gaussian $p_X^* = \mathcal{N}(0, 1)$.

persists even with perfect density models, and therefore goes beyond issues of estimation, approximation, or optimization errors (Bottou & Bousquet, 2008). We highlight that this issue is pervasive as it occurs even in low-dimensional settings and for a variety of density-based methods for anomaly detection.

## 2 DENSITY-BASED ANOMALY DETECTION

### 2.1 UNSUPERVISED ANOMALY DETECTION: PROBLEM STATEMENT

Unsupervised anomaly detection is a classification problem (Moya et al., 1993; Schölkopf et al., 2001), where one aims at distinguishing between regular points (*inliers*) and irregular points (*outliers*). However, as opposed to the usual classification task, labels distinguishing inliers and outliers are not provided for training, if outliers are even provided at all. Given a input space $\mathcal{X} \subseteq \mathbb{R}^D$, the task can be summarized as partitioning this space between the subset of outliers $\mathcal{X}_{out}$ and the subset of inliers $\mathcal{X}_{in}$, i.e., $\mathcal{X}_{out} \cup \mathcal{X}_{in} = \mathcal{X}$ and $\mathcal{X}_{out} \cap \mathcal{X}_{in} = \varnothing$. When the training data is distributed according to the probability measure $P_X^*$ (with density $p_X^*$ [1]), one would usually pick the set of regular points $\mathcal{X}_{in}$ such that this set contains the majority (but not all) of the mass (e.g., 95%) of this distribution, i.e., $P_X^*(\mathcal{X}_{in}) = 1 - \alpha \in \left(\frac{1}{2}, 1\right)$. But, for any given $\alpha$, there exists in theory an infinity of corresponding partitions into $\mathcal{X}_{in}$ and $\mathcal{X}_{out}$ (see Figure 1). How are these partitions defined to match our intuition of inliers and outliers? We will focus in this paper on recently used methods based on probability density.

### 2.2 DENSITY SCORING

When talking about outliers, infrequent observations, the association with probability can be quite intuitive. For instance, one would expect an anomaly to happen rarely and be unlikely. Since the language of statistics often associate the term *likelihood* with quantities like $p_X^{(\theta)}(x)$, one might consider an unlikely sample to have a low "likelihood", that is a low probability density $p_X^*(x)$. Conversely, regular samples would have a high density $p_X^*(x)$ following that reasoning. This is an intuition that is not only prevalent in several modern anomaly detection methods (Bishop, 1994; Blei et al., 2017; Hendrycks et al., 2018; Kirichenko et al., 2020; Rudolph et al., 2020; Liu et al., 2020) but also in techniques like low-temperature sampling (Graves, 2013) used for example in Kingma & Dhariwal (2018) and Parmar et al. (2018).

The associated approach, described in Bishop (1994), consists in defining the inliers as the points whose density exceed a certain threshold $\lambda > 0$ (for example, chosen such that inliers include a predefined amount of mass, e.g., 95%), making the modes the most regular points in this setting. $\mathcal{X}_{out}$ and $\mathcal{X}_{in}$ are then respectively the lower-level and upper-level sets $\{x \in \mathcal{X}, p_X^*(x) \leq \lambda\}$ and $\{x \in \mathcal{X}, p_X^*(x) > \lambda\}$ (see Figure 2b).

---

[1]We will also assume in the rest of the paper that for any $x \in \mathcal{X}, p_X^*(x) > 0$.

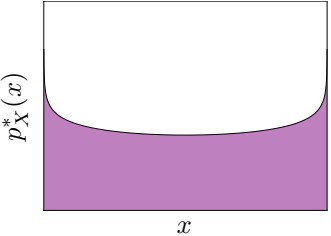 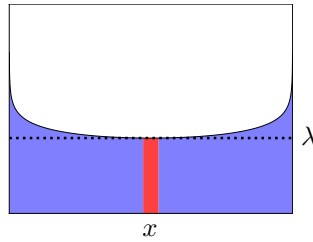 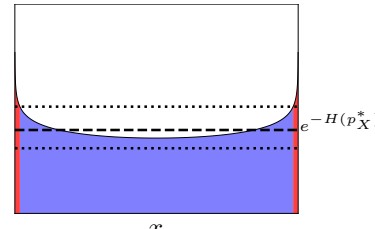

(a) An example of a distribution density $p_X^*$.

(b) Density scoring method applied to the distribution $p_X^*$.

(c) Typicality test method (with one sample) applied to the distribution $p_X^*$.

Figure 2: Illustration of different density-based methods applied to a particular one-dimensional distribution $p_X^*$. Outliers are in red and inliers are in blue. The thresholds are picked so that inliers include $95\%$ of the mass. In Figure 2b, inliers are considered as the points with density above the threshold $\lambda > 0$ while in Figure 2c, they are the points whose log-density are in the $\epsilon$-interval around the negentropy $-H(p_X^*)$.

## 2.3 TYPICALITY TEST

The *Gaussian Annulus theorem* (Blum et al., 2016) (generalized in Vershynin, 2019) attests that most of the mass of a high-dimensional standard Gaussian $\mathcal{N}(0, \mathbb{I}_D)$ is located close to the hypersphere of radius $\sqrt{D}$. However, the mode of its density is at the center $0$. A natural conclusion is that the *curse of dimensionality* creates a discrepancy between the density upper-level sets and what we expect as inliers (Choi et al., 2018; Nalisnick et al., 2019; Morningstar et al., 2020; Dieleman, 2020). This motivated Nalisnick et al. (2019) to propose another method for testing whether a point is an inlier or not, relying on a measure of its *typicality*. This method relies on the notion of *typical set* (Cover, 1999) defined by taking as inliers points whose average log-density is close to the average log-density of the distribution (see Figure 2c).

**Definition 1** (Cover, 1999). *Given independent and identically distributed elements $\left(x^{(n)}\right)_{n \leq N}$ from a distribution with density $p_X^*$, the typical set $A_\epsilon^{(N)}(p_X^*) \subset \mathcal{X}^N$ is made of all sequences that satisfy:*

$$\left| H(p_X^*) + \frac{1}{N} \sum_{n=1}^{N} \log p_X^* \left(x^{(n)}\right) \right| \leq \epsilon,$$

*where $H(X) = -\mathbb{E}[\log p_X^*(X)]$ is the (differential) entropy and $\epsilon > 0$ a constant.*

This method matches the intuition behind the Gaussian Annulus theorem on the set of inliers of a high-dimensional standard Gaussian. Indeed, using a concentration inequality, we can show that $\lim_{N \to +\infty} \left( P_{(X_i)_{1 \leq n \leq N}}^* \left( A_\epsilon^{(N)} \right) \right) = 1$, which means that with $N$ large enough, $A_\epsilon^{(N)}(p_X^*)$ will contain most of the mass of $(p_X^*)^N$, justifying the name *typicality*.

## 3 THE ROLE OF REPARAMETRIZATION

Given the anomaly detection problem formulation Subsection 2.1, we are interested in reasoning about the properties a solution ought to satisfy, in the ideal case of infinite data and capacity. For density-based methods this means that $p_X^{(\theta)} = p_X^*$. This setting is appealing as it gives space for theoretical results without worrying about the underfitting or overfitting issues mentioned by Hendrycks et al. (2018); Fetaya et al. (2020); Morningstar et al. (2020); Kirichenko et al. (2020); Zhang et al. (2020).

Although we work in practice on points (e.g., vectors), it is important to keep in mind that these points are actually representations of an underlying outcome. As a random variable, $X$ is by definition the function from this outcome $\omega$ to the corresponding observation $x = X(\omega)$. However, at its core, an anomaly detection solution aims at classifying outcomes through these measurements. How is the

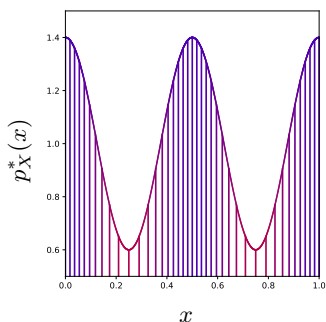 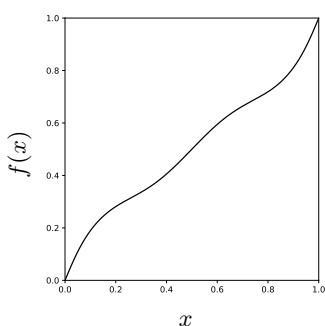 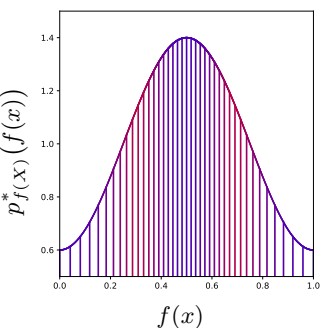

(a) An example of a distribution density $p_X^*$.

(b) Example of an invertible function $f$ from $[0, 1]$ to $[0, 1]$.

(c) Resulting density $p_{f(X)}^*$ from applying $f$ to $X \sim p_X^*$ as a function of the new axis $f(x)$.

Figure 3: Illustration of the change of variables formula and how much the application of a bijection can affect the density of the points considered in a one-dimensional case. In Figures 3a and 3c, points $x$ with high density $p_X^*(x)$ are in blue and points with low density $p_X^*(x)$ are in red.

choice of $X$ affecting the problem of anomaly detection? While several papers studied the effects of a change of representation through the lens of inductive bias (Kirichenko et al., 2020; Zhang et al., 2020), we investigate the more fundamental effects of reparametrizations $f$. To sidestep concerns about loss of information (Winkens et al., 2020), we study the particular case of an invertible map $f$.

The measurements $x = X(\omega)$ and $f(x) = (f \circ X)(\omega)$ represent the same outcome $\omega$ (although differently), and, since $x$ and $f(x)$ are connected by an invertible transformation $f$, the same method applied respectively to $X$ or $f(X)$ should classify them with the same label, either as an inlier or an outlier. The target of these methods is to essentially assess the regularity of the outcome $\omega$. From this, we could ideally make the following requirement for a solution to anomaly detection.

**Principle.** *In an infinite data and capacity setting, the result of an anomaly detection method should be invariant to any* continuous *invertible reparametrization $f$.*

Do density-based methods follow this principle? To answer that question, we look into how density behaves under a reversible change of representation. In particular, the change of variables formula (Kaplan, 1952) (used in Tabak & Turner, 2013; Dinh et al., 2014; Rezende & Mohamed, 2015), formalizes a simple intuition of this behavior: where points are brought closer together the density increases whereas this density decreases when points are spread apart. The formula itself is written as:

$$p_{f(X)}^*\big(f(x)\big) = p_X^*(x) \left| \frac{\partial f}{\partial x^T}(x) \right|^{-1} ,$$

where $\left| \frac{\partial f}{\partial x^T}(x) \right|$ is the Jacobian determinant of $f$ at $x$, a quantity that reflects a local change in volume incurred by $f$. Figure 3 already illustrates how the function $f$ (Figure 3b) can spread apart points close to the extremities to decrease the corresponding density round 0 and 1, and, as a result, turns the density on the left (Figure 3a) into the density on the right (Figure 3c). With this example, one can wonder to which degree an invertible change of representation can affect the density and the anomaly detection methods presented in Subsections 2.2 and 2.3 that use it.

## 4 LEVERAGING THE CHANGE OF VARIABLES FORMULA

### 4.1 UNIFORMIZATION

We start by showing that unambiguously defining outliers and inliers with any density-based approach becomes impossible when considering a particular type of invertible reparametrization of the problem, irrespective of dimensionality.

Under weak assumptions, one can map any distribution to a uniform distribution using an invertible transformation (Hyvärinen & Pajunen, 1999). This is in fact a common strategy for sampling from

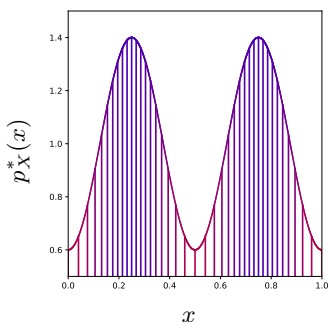 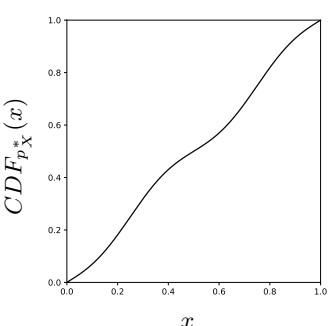 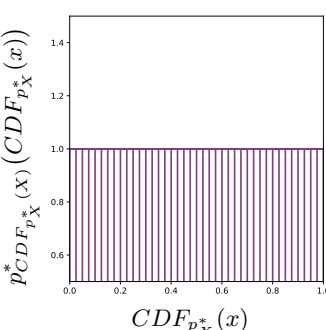

(a) An example of a distribution density $p_X^*$. Points $x$ with high density $p_X^*(x)$ are in blue and points with low density $p_X^*(x)$ are in red.

(b) The corresponding cumulative distribution function $CDF_{p_X^*}$.

(c) The resulting density from applying $CDF_{p_X^*}$ to $X \sim p_X^*$ is $p_{CDF_{p_X^*}(X)}^* = \mathcal{U}([0,1])$, therefore we color all the points the same.

Figure 4: Illustration of the one-dimensional case version of a Knothe-Rosenblatt rearrangement, which is just the application of the cumulative distribution function $CDF_{p_X^*}$ on the variable $x$.

complicated one-dimensional distributions (Devroye, 1986). Figure 4 shows an example of this where a bimodal distribution (Figure 4a) is pushed through an invertible map (Figure 4b) to obtain a uniform distribution (Figure 4c).

To construct this invertible uniformization function, we rely on the notion of Knothe-Rosenblatt rearrangement (Rosenblatt, 1952; Knothe et al., 1957). A Knothe-Rosenblatt rearrangement (notably used in Hyvärinen & Pajunen, 1999) is defined for a random variable $X$ distributed according to a strictly positive density $p_X^*$ with a convex support $\mathcal{X}$, as a continuous invertible map $f^{(KR)}$ from $\mathcal{X}$ onto $[0,1]^D$ such that $f^{(KR)}(X)$ follows a uniform distribution in this hypercube. This rearrangement is constructed as follows: $\forall d \in \{1,...,D\}, f^{(KR)}(x) = CDF_{p_{X_d|X_{<d}}^*}(x_d \mid x_{<d})$ where $CDF_p$ is the cumulative distribution function corresponding to the density $p$.

In these new coordinates, neither the density scoring method nor the typicality test approach can discriminate between inliers and outliers in this uniform $D$-dimensional hypercube $[0,1]^D$. Since the resulting density $p_{f^{(KR)}(X)}^* = 1$ is constant, the density scoring method attributes the same regularity to every point. Moreover, a typicality test on $f^{(KR)}(X)$ will always succeed as

$$\forall \epsilon > 0, N \in \mathbb{N}^*, \forall \left(x^{(n)}\right)_{n \leq N}, \left| H\left(p_{f^{(KR)}(X)}^*\right) + \frac{1}{N}\sum_{n=1}^{N} \log p_{f^{(KR)}(X)}^* \left(f^{(KR)}\left(x^{(n)}\right)\right)\right|$$

$$= \left| H\left(\mathcal{U}\left([0,1]^D\right)\right) + \frac{1}{N}\sum_{n=1}^{N} \log(1)\right| = 0 \leq \epsilon.$$

However, these uniformly distributed points are merely a different representation of the same initial points. Therefore, if the identity of the outliers is ambiguous in this uniform distribution, then anomaly detection in general should be as difficult.

## 4.2 ARBITRARY SCORING

While a particular parametrization can prevent density-based outlier detection methods from separating between outliers and inliers, we find that it is also possible to build a reparametrization of the problem to impose to each point an arbitrary density level in the new representation. To illustrate this idea, consider some points from a distribution whose density is depicted in Figure 5a and a score function indicated in red in Figure 5b. In this example, high-density regions correspond to areas with low score value (and vice-versa). We show that there exists a reparametrization (depicted in Figure 5c) such that the density in this new representation (Figure 5d) now matches the desired score, which can be designed to mislead density-based methods into a wrong classification of anomalies.

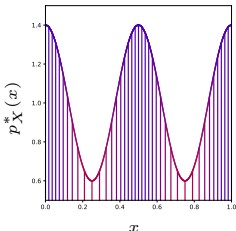
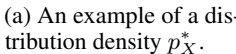
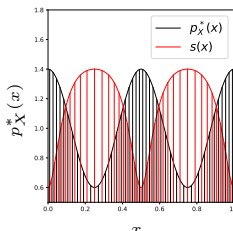
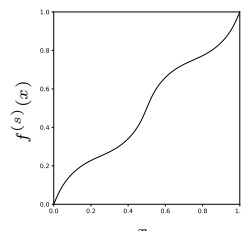
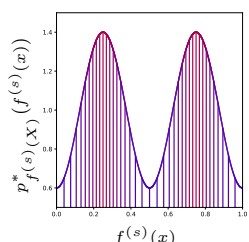

(a) An example of a distribution density $p_X^*$.

(b) The distribution $p_X^*$ (in black) and the desired density scoring $s$ (in red).

(c) A continuous invertible reparametrization $f^{(s)}$ such that $p_{f^{(s)}(X)}^*\big(f^{(s)}(x)\big) = s(x)$.

(d) Resulting density $p_{f^{(s)}(X)}^*$ from applying $f^{(s)}$ to $X \sim p_X^*$ as a function of $f^{(s)}(x)$.

Figure 5: Illustration of how we can modify the space with an invertible function so that each point $x$ follows a predefined score. In Figures 5a and 5d, points $x$ with high density $p_X^*(x)$ are in blue and points with low density $p_X^*(x)$ are in red.

**Proposition 1.** *For any variable $X \sim p_X^*$ with $p_X^*$ continuous strictly positive (with $\mathcal{X}$ convex) and any measurable continuous function $s : \mathcal{X} \to \mathbb{R}_+^*$ bounded below by a strictly positive number, there exists a continuous bijection $f^{(s)}$ such that for any $x \in \mathcal{X}, p_{f^{(s)}(X)}\big(f^{(s)}(x)\big) = s(x)$ almost everywhere.*

*Proof.* We write $x$ to denote $(x_1, \ldots, x_{D-1}, x_D)$ and $(x_{<D}, t)$ for $(x_1, \ldots, x_{D-1}, t)$. Let $f^{(s)} : \mathcal{X} \to \mathcal{Z} \subset \mathbb{R}^D$ be a function such that

$$\big(f^{(s)}(x)\big)_D = \int_0^{x_D} \frac{p_X^*\big((x_{<D}, t)\big)}{s\big((x_{<D}, t)\big)} \, dt,$$

and $\forall d \in \{1, ..., D-1\}, \big(f^{(s)}(x)\big)_d = x_d$. As $s$ is bounded below, $f^{(s)}$ is well defined and invertible. By the change of variables formula,

$$\forall x \in \mathcal{X}, \ p_{f^{(s)}(X)}^*\big(f^{(s)}(x)\big) = p_X^*(x) \cdot \left|\frac{\partial f^{(s)}}{\partial x^T}(x)\right|^{-1} = p_X^*(x) \cdot \left(\frac{p_X^*(x)}{s(x)}\right)^{-1} = s(x).$$

$\square$

If $\mathcal{X}_{in}$ and $\mathcal{X}_{out}$ are respectively the true sets of inliers and outliers, we can pick a ball $A \subset \mathcal{X}_{in}$ such that $P_X^*(A) = \alpha < 0.5$, we can choose $s$ such that for any $x \in (\mathcal{X} \setminus A), s(x) = 1$ and for any $x \in A, s(x) = 0.1$. With this choice of $s$ (or a smooth approximation) and the function $f^{(s)}$ defined earlier, both the density scoring and the (one-sample) typical set methods will consider the set of inliers to be $(\mathcal{X} \setminus A)$ while $\mathcal{X}_{out} \subset (\mathcal{X} \setminus A)$, making their results completely wrong. While we can also reparametrize the problem so that these methods may succeed, such reparametrization requires knowledge of $(p_X^*/s)(x)$. Without any constraints on the space considered, individual densities can be arbitrarily manipulated, which reveals how little these quantities say about the underlying outcome in general.

## 4.3    CANONICAL DISTRIBUTION

Since our analysis in Subsections 4.1 and 4.2 reveals that densities or low typicality regions are not sufficient conditions for an observation to be an anomaly, whatever its distribution or its dimension, we are now interested in investigating whether additional realistic assumptions can lead to some guarantees for anomaly detection. Motivated by several representation learning algorithms which attempt to learn a mapping to a predefined distribution (e.g., a standard Gaussian, see Chen & Gopinath, 2001; Kingma & Welling, 2014; Rezende et al., 2014; Dinh et al., 2014; Krusinga et al., 2019) we consider the more restricted setting of a fixed distribution of our choice, whose regular regions could for instance be known. Surprisingly, we find that it is possible to exchange the densities of an inlier and an outlier even within a canonical distribution.

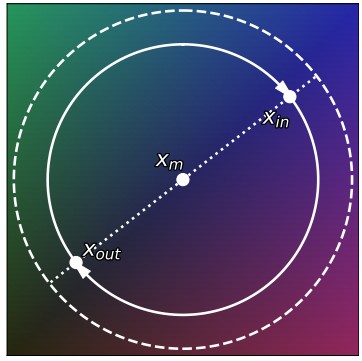 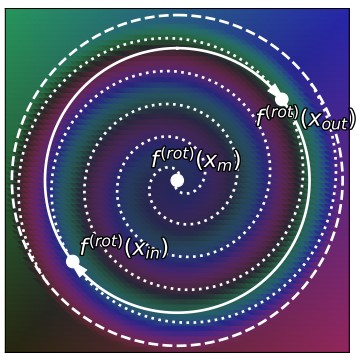

(a) Points $x_{in}$ and $x_{out}$ in a uniformly distributed subset. $f^{(rot)}$ will pick a two-dimensional plane and use the polar coordinate using the mean $x_m$ of $x_{in}$ and $x_{out}$ as the center.

(b) Applying a bijection $f^{(rot)}$ exchanging the points $x_{in}$ and $x_{out}$. $f^{(rot)}$ is a rotation depending on the distance from the mean $x_m$ of $x_{in}$ and $x_{out}$ in the previously selected two-dimensional plane.

Figure 6: Illustration of the norm-dependent rotation, a locally-acting bijection that allows us to swap two different points while preserving a uniform distribution (as a volume-preserving function).

**Proposition 2.** *For any strictly positive density function $p_X^*$ over a convex space $\mathcal{X} \subseteq \mathbb{R}^D$ with $D > 2$, for any $x_{in}, x_{out}$ in the interior $\mathcal{X}^\circ$ of $\mathcal{X}$, there exists a continuous bijection $f : \mathcal{X} \to \mathcal{X}$ such that $p_X^* = p_{f(X)}^*$, $p_{f(X)}^* \left( f\left(x^{(in)}\right)\right) = p_X^* \left(x^{(out)}\right)$, and $p_{f(X)}^* \left( f\left(x^{(out)}\right)\right) = p_X^* \left(x^{(in)}\right)$.*

We provide a sketch of proof and put the details in Appendix A. We rely on the transformation depicted in Figure 6, which can swap two points while acting in a very local area. If the distribution of points is uniform inside this local area, then this distribution will be unaffected by this transformation. In order to arrive at this situation, we use the uniformization method presented in Subsection 4.1, along with a linear function to fit this local area inside the support of the distribution (see Figure 7). Once those two points have been swapped, we can reverse the functions preceding this swap to recover the original distribution overall.

Since the resulting distribution $p_{f(X)}^*$ is identical to the original $f_X^*$, then their entropies are the same $H\left(p_{f(X)}^*\right) = H\left(f_X^*\right)$. Hence, when $x_{in}$ and $x_{out}$ are respectively an inlier and an outlier, whether in terms of density scoring or typicality, there exists a reparametrization of the problem conserving the overall distribution while still exchanging their status as inlier/outlier. We provide an example applied to a standard Gaussian distribution in Figure 8.

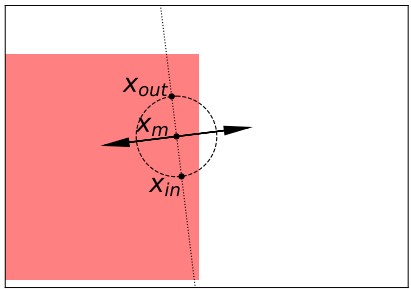 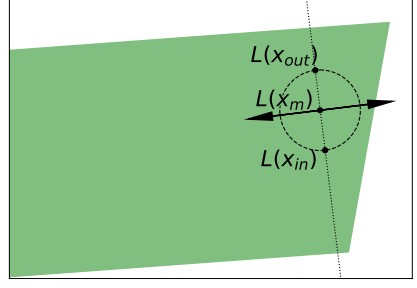

(a) When taking two points $x_{in}$ and $x_{out}$ inside the hypercube $[0,1]^D$, there is sometimes no $L_2$-ball centered in their mean $x_m$ containing both $x_{in}$ and $x_{out}$.

(b) However, given $x_{in}$ and $x_{out}$, one can apply an invertible linear transformation $L$ such that there exists a $L_2$-ball centered in their new mean $L(x_m)$ containing both $L(x_{in})$ and $L(x_{out})$. If the distribution was uniform inside $[0,1]^D$, then it is now also uniform inside $L\left([0,1]^D\right)$

Figure 7: We illustrate how, given $x_{in}$ and $x_{out}$ in a uniformly distributed hypercube $[0,1]^D$, one can modify the space such that $f^{(rot)}$ shown in Figure 6 can be applied without modifying the distribution.

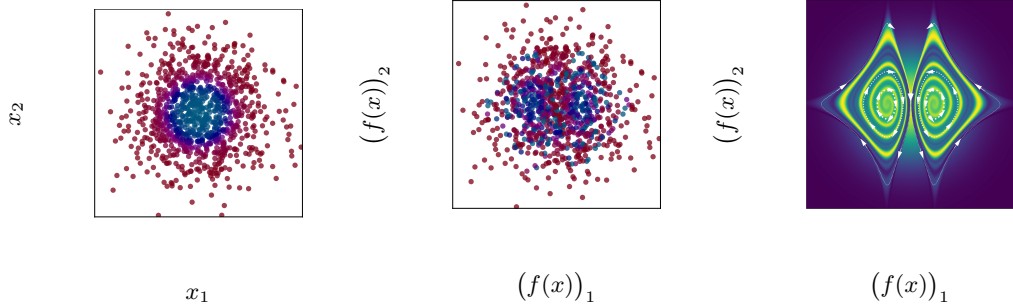

(a) Points sampled from $p_X^*$ = $\mathcal{N}(0, \mathbb{I}_2)$.

(b) Applying a bijection $f$ that preserves the distribution $p_{f(X)}^*$ = $\mathcal{N}(0, \mathbb{I}_2)$ to the points in Figure 8a.

(c) The original distribution $p_X^*$ with respect to the new coordinates $f(x)$, $p_X^* \circ f^{-1}$.

Figure 8: Application of a transformation using the bijection in Figure 6 to a standard Gaussian distribution $\mathcal{N}(0, \mathbb{I}_2)$, leaving it overall invariant.

This result is important from a representation learning perspective and a complement to the general non-identifiability result in several representation learning approaches (Hyvärinen & Pajunen, 1999; Locatello et al., 2019). It means that learning a representation with a predefined, well-known distribution and knowing the true density $p_X^*$ are not sufficient conditions to control the individual density of each point and accurately distinguish outliers from inliers.

## 5 DISCUSSION

Fundamentally, density-based methods for anomaly detection rely on the belief that density, as a quantity, conveys useful information to assess whether an outcome is an outlier or not. For example, several density-based methods operate in practice on features learned independently from the anomaly detection task (Lee et al., 2018; Krusinga et al., 2019; Morningstar et al., 2020; Winkens et al., 2020) or on the original input features (Nalisnick et al., 2018; Hendrycks et al., 2018; Kirichenko et al., 2020; Rudolph et al., 2020; Nalisnick et al., 2019). In general, there is no evidence that the density in these representations will carry any useful information for anomaly detection bringing into question whether performance of probabilistic models on this task (e.g., Du & Mordatch, 2019; Grathwohl et al., 2019; Kirichenko et al., 2020; Liu & Abbeel, 2020) reflects goodness-of-fit of the density model. On the contrary, we have proven in this paper that density-based anomaly detection methods are inconsistent across a range of possible representations [2], even under strong constraints on the distribution, which suggests that finding the right input representation for meaningful density-based anomaly detection requires privileged information, as discussed in Subsection 4.2. Moreover, several papers have pointed to existing problems in commonly used input representations; for example, the geometry of a bitmap representation does not follow our intuition of semantic distance (Theis et al., 2016), or images can come from photographic sensors tuned to specific populations (Roth, 2009; Buolamwini & Gebru, 2018). This shows how strong of an otherwise understated assumption it is to suppose that the methods presented in Subsection 2.2 and Subsection 2.3 would work on input representations. This is particularly problematic for applications as critical as autonomous vehicle navigation or medical decision-making.

While defining anomalies might be impossible without prior knowledge (Winkens et al., 2020) as out-of-distribution detection is an ill-posed problem (Choi et al., 2018; Nalisnick et al., 2019; Morningstar et al., 2020), several approaches make these assumptions more explicit. For instance, the density scoring method has also been interpreted in Bishop (1994) as a likelihood ratio method (Ren et al., 2019; Serrà et al., 2020; Schirrmeister et al., 2020), which, on the one hand, relies heavily on the definition of an arbitrary reference density as a denominator of this ratio but, on the other hand, is invariant to reparametrization. Inspired by the Bayesian approach from Choi et al. (2018), one can work on defining a prior distribution on possible reparametrizations over which to average (similarly to Jørgensen & Hauberg, 2020).

---

[2]Alternatively, this can be seen as a change of base distribution used to define a probability density as a Radon-Nikodym derivative.

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

## A  PROOF OF PROPOSITION 2

**Proposition 3.** *For any strictly positive density function $p_X^*$ over a convex space $\mathcal{X} \subseteq \mathbb{R}^D$ with $D > 2$, for any $x_{in}, x_{out}$ in the interior $\mathcal{X}^\circ$ of $\mathcal{X}$, there exists a continuous bijection $f : \mathcal{X} \to \mathcal{X}$ such that $p_X^* = p_{f(X)}^*$, $p_{f(X)}^* \left( f\left(x^{(in)}\right)\right) = p_X^* \left(x^{(out)}\right)$, and $p_{f(X)}^* \left( f\left(x^{(out)}\right)\right) = p_X^* \left(x^{(in)}\right)$.*

*Proof.* Our proof will rely on the following non-rigid rotation $f^{(rot)}$. Working in a hyperspherical coordinate system consisting of a radial coordinate $r > 0$ and $(D-1)$ angular coordinates $(\phi_i)_{i<D}$,

$$\forall d < D, \; x_d = r \left( \prod_{i=1}^{d-1} \sin(\phi_i) \right) \cos(\phi_d)$$

$$x_D = r \left( \prod_{i=1}^{D-2} \sin(\phi_i) \right) \sin(\phi_{D-1}),$$

where for all $i \in \{1, 2, ..., D-2\}$, $\phi_i \in [0, \pi)$ and $\phi_{D-1} \in [0, 2\pi)$, given $r_{max} > r_0 > 0$, we define the continuous mapping $f^{(rot)}$ as:

$$f^{(rot)} \left( (r, \phi_1, \ldots, \phi_{D-2}, \phi_{D-1}) \right) = \left( r, \phi_1, \ldots, \phi_{D-2}, \phi_{D-1} + \pi \frac{(r_{max} - r)_+}{r_{max} - r_0} [\text{mod } 2\pi] \right).$$

where $(\cdot)_+ = \max(\cdot, 0)$. This mapping only affects points inside $\mathcal{B}_2(0, r_{max})$, and exchanges two points corresponding to $(r_0, \phi_1, \ldots, \phi_{D-2}, \phi_{D-1})$ and $(r_0, \phi_1, \ldots, \phi_{D-2}, \phi_{D-1} + \pi)$ in a continous way (see Figure 6). Since the Jacobian determinant of the hyperspherical coordinates transformation is not a function of $\phi_{D-1}$, $f^{(rot)}$ is volume-preserving in cartesian coordinates.

Let $f^{(KR)}$ be a Knothe-Rosenblatt rearrangement of $p_X^*$, $f^{(KR)}(X)$ is uniformly distributed in $[0, 1]^D$. Let $z^{(in)} = f^{(KR)}\left(x^{(in)}\right)$ and $z^{(out)} = f^{(KR)}\left(x^{(out)}\right)$. Since $f^{(KR)}$ is continuous, $z^{(in)}, z^{(out)}$ are in the interior $(0, 1)^D$. Therefore, there is an $\epsilon > 0$ such that the $L_2$-balls $\mathcal{B}_2\left(z^{(in)}, \epsilon\right)$ and $\mathcal{B}_2\left(z^{(out)}, \epsilon\right)$ are inside $(0, 1)^D$. Since $(0, 1)^D$ is convex, so is their convex hull.

Let $r_0 = \frac{1}{2} \left\| z^{(in)} - z^{(out)} \right\|_2$ and $r_{max} = r_0 + \epsilon$. Given $z \in (0, 1)^D$, we write $z_\|$ and $z_\perp$ to denote its parallel and orthogonal components with respect to $\left(z^{(in)} - z^{(out)}\right)$. We consider the linear bijection $L$ defined by

$$L(z) = z_\| + \epsilon^{-1} r_{max} z_\perp.$$

Let $f^{(z)} = L \circ f^{(KR)}$. Since $L$ is a linear function (i.e., with constant Jacobian), $f^{(z)}(X)$ is uniformly distributed inside $L\left([0, 1]^D\right)$. If $z^{(m)}$ is the mean of $z^{(in)}$ and $z^{(out)}$, then $f^{(z)}(\mathcal{X})$ contains $\mathcal{B}_2\left(L\left(z^{(m)}\right), r_{max}\right)$ (see Figure 7). We can then apply the non-rigid rotation $f^{(rot)}$ defined earlier, centered on $L\left(z^{(m)}\right)$ to exchange $L\left(z^{(in)}\right)$ and $L\left(z^{(out)}\right)$ while maintaining this uniform distribution.

We can then apply the bijection $\left(f^{(z)}\right)^{-1}$ to obtain the invertible map $f = \left(f^{(z)}\right)^{-1} \circ f^{(rot)} \circ f^{(z)}$ such that $p_{f(X)}^* = f_X^*$, $p_{f(X)}^* \left( f\left(x^{(in)}\right)\right) = p_X^* \left(x^{(out)}\right)$, and $p_{f(X)}^* \left( f\left(x^{(out)}\right)\right) = p_X^* \left(x^{(in)}\right)$. $\qquad \square$

