# OpenReview forum: "Perfect density models cannot guarantee anomaly detection"
_ICLR.cc/2021/Conference — Reject_

### Official Review · AnonReviewer4 · 2020-10-26
**A well-written paper; unclear about the reasonableness of the main principle**

**Rating:** 4
**Confidence:** 3

**Review:**

**Update**

My impression after the extensive discussion is that the remaining differences are possibly too subjective to come to an agreement:

1) Whether the fact that the invertible reparameterization principle does not hold for anomaly detection represents a significant theoretical contribution. To me, I still don’t quite see why, but I do believe there are other researchers with a stronger theory background more qualified than me to evaluate this, I will therefore downgrade my experience score by 1.
2) Whether the fact that the invertible reparameterization principle does not hold has promise to further understand the practical issues like the CIFAR10 vs SVHN phenomenon beyond things that have already been discussed in the literature - I think quite likely not, the different arguments for and against have been discussed in our back and forth I think.

I invite the AC to go through the discussion here to see the detailed arguments/rebuttals.

My personal impression, in case the authors are interested, is also the manuscript might be further improved by giving more space to these two questions, i.e. the motivation of the principle and instead moving some of the proofs to the supplementary. I think it is quite straightforward to understand that the principle cannot hold and more difficult to understand why it should hold in the first place.


**Summary**
In this manuscript, the authors highlight one phenomenon about density-based anomaly detection: Even the perfect density model can assign arbitrary densities if you allow an arbitrary invertible reparametrization of your data before estimating the densities and therefore will have arbitrary inlier/outlier classification. Therefore, methods that assign anomalies purely based on low density values cannot achieve a principle proposed by the authors: “anomaly detection methods should be invariant to any invertible reparametrization f”.

**Main Impression**
The manuscript is well-structured and well-written, allowing me to quickly understand the main argument of the paper (I hope), thanks for that. I found the analyses to be technically correct to the level I was able to analyze them.

Overall, I have some doubts about how much new practically relevant things can be learned about anomaly detection from the argument laid out in the paper.

**Details about Main Criticism**
1) The fact that densities can be arbitrarily changed by invertible functions is already an important building block of normalizing flows  - in order to learn any distribution starting from a predefined prior like a gaussian distribution. So in itself it does not seem like a new aspect to me.
2) The effect of invertible reparametrizations on anomaly detection has already surfaced in the literature: E.g., in Nalisnick, 2018, they already report very different anomaly detection results if one uses only p(z) and ignores the log-determinant for an affine Glow Network. This setup there, as far as I see, fits in the framework here, with f there implemented by the affine Glow network.
3) I am not sure I agree with the principle that anomaly detection methods should be invariant to invertible reparametrizations. An example: Let’s say you visualize the samples from a gaussian distribution as grayscale values, with gray at zero, and positive values becoming brighter, negative values becoming darker. Then very dark values as well as very bright values would be visually outliers. Now imagine you do an invertible reparametrization such that the distribution becomes bimodal with the modes further away from the center. Visually, originally somewhat darker/brighter values would be stretched to contain a lot of very dark/very bright values. Now some very dark values/very bright values would no longer be outliers in the perfect density model, but I would argue they would also visually no longer be outliers.
4) Further, the original parametrization of e.g., the way images are stored is not arbitrary, but a parametrization where we as humans can perceive the content.
5) Note also that in e.g., normalizing flows literature, it is always attempted that the estimated density is a lower bound for the discrete probabilities. There are also models directly modelling the discrete probabilities, e.g., see https://arxiv.org/abs/1905.07376 https://arxiv.org/abs/2006.12459. Note that normally, for density-based models one can compare the continuous and discrete models, while the proposed reparametrization logic here is not applicable for discrete models at all.
6) The logic also rests on the assumption of the “true” density always being not zero/positive (footnote page 2). For the phenomena the paper references, I am not sure this is a good assumption. E.g., should SVHN-like zeros really have positive density under the true density of CIFAR-like frogs? One could also argue they have different support?
7) In my view, for the case of images, the anomaly detection phenomena are already well-explained by the line of thoughts in Ren et al., 2019; Serrà et al., 2020; Schirrmeister et al., 2020 and Kirichenko et al., 2020 so I am not sure in which scenarios the invertible-reparametrization aspect adds further understanding.

To summarize again, my main problem is, I am not convinced that “the result of an anomaly detection method should be invariant to any invertible reparametrization f” and that the failure to do so by density-based methods helps understand their failures as pointed out in recent literature.

**Suggestion for Improvement**
What would change my mind about the value of the phenomenon highlighted here is if there are examples in the literature where people use densities that are strongly affected by a reparametrization and are not aware of it. I think such examples would also be more useful rather than showing it also works with canonical distribution etc. I think the entire principle with invertible reparameterizations is clear enough, and any analyses showing it works in many mathematical settings, could be moved to supplementary. The more important question to me is when this principle makes sense at all.


**Other Points**
Is the whole setting in 2.1 one also used in literature? References might be helpful for “one would usually pick the set of regular points X_in such that this set contains the majority (but not all) of the mass (e.g.,95%) of this distribution”

“Without any constraints on the space considered, individual densities can be arbitrarily manipulated, which reveals how little these quantities mean in general.”
-> I don’t agree this shows how little densities mean in general. I think it shows how little they mean once you allow arbitrary invertible reparametrizations, and that maybe allowing this is not a good idea ;)

“It means that learning a representation with a predefined,  well-known distribution and knowing the true density p∗(X) are not sufficient conditions to control the individual density of each point and accurately distinguish outliers from inliers.”
-> I think this statement is a bit strange. If you know the true density at original parametrization and have access to the data in original parameterization, you can evaluate it and accurately distinguish outlier/inlier if they are defined by density-levels there in the first place. Also if they are invertibly transformed and you know the invertible transformation you can also invert it… if they are not defined by density-level at original parametrization, then the question is what are they defined by anyways, and you have a whole new question in my view.

I encourage the authors to also clear up any misunderstandings of the current manuscript from my side.

---

> ### Author Response · Authors · 2020-11-16
> **Addressing your points (1/2)**
>
> Dear reviewer 4,
> Thank you very much for sharing your concerns and developing your point of view. We are happy to attempt to address your points individually:
> - (1) “The fact that densities can be arbitrarily changed by invertible functions is already an important building block of normalizing flows”
> We do not wish to claim the novelty of using the change of variables formula for probability density and cited the adequate work to dispel any confusion on that. Are there still part of the paper that would suggest otherwise?
> - (2) “The effect of invertible reparametrizations on anomaly detection has already surfaced in the literature: E.g., in [Nalisnick, 2018](https://arxiv.org/abs/1810.09136), they already report very different anomaly detection results if one uses only p(z)”
> We have indeed cited Nalisnick. They indeed report results using only p(z) in that paper. However, while their empirical observations completely entangle the practical aspect of deep generative models fitting, including inductive bias and under/overfitting, and more fundamental problems, such as the ones we highlight in this submission. Their conclusion was written as such: “[w]hile we cannot conclude that this is a pathology in deep generative models, it does suggest the need for further work on generative models and their evaluation”. We are more definitive on the fact that the expectation of a density model with respect to anomaly detection is a strong and can be unfounded.
> - (3) “Now imagine you do an invertible reparametrization such that the distribution becomes bimodal with the modes further away from the center. Visually, originally somewhat darker/brighter values would be stretched to contain a lot of very dark/very bright values. Now some very dark values/very bright values would no longer be outliers in the perfect density model, but I would argue they would also visually no longer be outliers”
> First, we want to make it clear that a change of representation does not mean in any way that the original input should change. You can think of it as encryption, an encrypted grey value should still be seen as a grey value, you just need to decode it.
> - (3 & 4 & 5 & 7) We also want to provide a counterexample to the intuition behind your example. In a bitmap representation. Let’s take a one pixel image for example to not only fit the bitmap representation narrative (point 4) but also to eliminate the hypothesis of texture bias inside convolutional models as a cause of failure (point 7). Let’s consider the following mixture distribution: $\mathcal{U}([255, 256]^3) $(white) with mixture probability 0.01 and $\mathcal{U}([0, 10]^3)$ (shades of black) with mixture probability 0.99. If we put every sample side to side in one image, we obtain the following image (https://imgur.com/a/bNOvBJl), where one would consider the white colored pixels to be outliers (point 3). However, these white values have a density of 0.01 each while darker values have a density of 0.00099 (<0.01) each. The same goes for probabilities in a $\\{0, …, 255\\}$ discretization of this counter-example (point 5). In this bitmap representation, the density values go against the intuition of associating density values with anomaly detection, despite knowing the underlying density model perfectly (therefore no inductive bias involved).
> - (6) This is true that this can be seen as a strong assumption but strictly positive densities are dense in the space of continuous densities in the strong topology, moreover this assumption allows for a proof that is easier to follow for the reader.
> - (7) “In my view, for the case of images, the anomaly detection phenomena are already well-explained by the line of thoughts in Ren et al., 2019; Serrà et al., 2020; Schirrmeister et al., 2020 and Kirichenko et al., 2020”
> To clarify, we assume that you mean that in the case of images, the texture bias that convolutional models have is a sufficient explanation for failure of density-based anomaly detection. At least [Kirichenko et al. (2020)](https://arxiv.org/abs/2006.08545) mention by name “inductive bias” which suggest to us that, in this narrative, this failure of density models for anomaly detection would be solved if a perfect density model was provided. We show in general (beyond the image case) that it’s not true. Moreover, we hope that the one pixel counter example we provided was convincing enough to show that this narrative can be insufficient even in the image case (albeit a simple one).

---

> > ### Comment · AnonReviewer4 · 2020-11-16
> > **Paper should argue more clearly when density-based anomaly-*definition* is unreasonable**
> >
> > Thanks for your detailed explanations and your insightful example.
> >
> > I will try to develop my viewpoint a bit more clearly, even where we might already agree, so we can discuss it further:
> >
> > To me, the unsupervised anomaly detection problem has at least two parts:
> > 1) The definition of anomalies, what are anomalies?
> > 2) Given the definition, how to detect the anomalies given samples of the distribution
> >
> > One issue to me is that the principle that density-based anomaly detection should be invariant to any invertible reparametrization mixes 1 and 2 together. If one a priori defines anomalies as points as points with low probability density, then after invertible reparametrization, which points are anomalous *should* change as we have a new distribution and densities have changed. This is the crucial difference to me to a classification problem where you have assigned fixed labels (from any criterion) and then any reparametrization does not change the label as the labels are fixed.
> >
> > What your example with the pixels shows to me:
> > Anomaly definition using densities does not always correspond to what we intuitively would define as outliers. Here I would argue one could also see it as an effect of the “model bias” of us/our visual system towards distinguishing less between grayscale values that are closely together. One way to extend  the example would be  the following invertible reparametrization (in discrete space): Map the white pixels 255 to 5, and pixels 5-254 to 6-255. So now the pixels with highest density are visually not outliers. Not as any contradiction, just to show again the example nicely shows: densities can, but do not necessarily correspond to what we intuitively would define as outliers.
> >
> > So the question to me seems whether to define anomalies by low probability densities in the first place. If one uses this definition, I do not agree the invertible reparametrization principle should hold. Note also that your pixel example to me shows that the definition of anomalies  is the real issue: In discrete space, there is no probability/density-changing invertible reparametrization, yet the problem persists.
> >
> > Whether density-based anomaly definition is sensible and whether there are better anomaly definitions corresponding more with human intution is certainly an interesting question. You provided an interesting simple example with a mismatch between intuition and density-based anomaly definition. It would be interesting if there are more realistic examples, especially if other anomaly definitions correspond more to intuition there. As you note, the references you provided do not completely do that as they cannot disentangle between model bias effects and potential effects of density-based definition. In other words, they cannot show whether the new representation has better anomaly definitions under a perfect density model or is more suited to detect anomalies under the specific model.
> >
> > Now to some individual points:
> >
> > (1) and (2) I did not get the impression that the paper claims to find that invertible models can transform densities arbitrarily. What I meant to say here is once one is aware that invertible models may transform densities arbitrarily, I would find it strange to expect anomaly definition based on densities should stay consistent after invertible reparameterizations. And using p(z) showed that anomaly definition can substantially change by invertible reparametrization.
> >
> > (6) Why I raise this is that  there is one case where any anomaly definition should agree: Points outside the support of the distribution should be anomalies. And this is also a point that is unaffected by any invertible reparametrization, as densities will remain 0.  Also, if you assume that some of the image class examples (SVHN-zero vs CIFAR10-frog or whatever) have different, non-overlapping support, the invertible reparametrization argument does not apply there.
> >
> > (7) Yes this is what I meant, exactly :) And yes I agree this line of view suggests that perfect density models trained in the original input space for the problems under consideration (SVHN vs. CIFAR) should work correctly (and this is in line with the argument in (6) here).
> >
> > I wrote a bit more now to hopefully more clearly define my viewpoint, even in points where we might already agree, also wrt to https://openreview.net/forum?id=MkrAyYVmt7b&noteId=HxfVX36hwJZ
> >
> > So maybe to state more what I still wonder:
> > 1) Do you agree there is little reason to think that density-based definition of anomalies explains the failure to distinguish e.g., SVHN and CIFAR10, or the other typical examples. In other words, do you think it is reasonable to expect that there is a chance the perfect density model of CIFAR10  will not define SVHN as anomalies?
> > 2) Do you think in general, there are real-world examples where density-based definition causes anomaly definition that contradicts our intuition?
> > 3) Do you think there are other anomaly definitions that circumvent this problem?

---

> > > ### Author Response · Authors · 2020-11-18
> > > **Knowing in advance when density-based anomaly-*definition* is reasonable requires privileged information**
> > >
> > > First of all, we’re happy to see that this submission is starting a fundamental and necessary discussion on how to define anomalies.
> > > Density-based approach attempts to flatten this complex problem of defining anomalies to density estimation. We do agree that reparametrization changes the density of points and therefore the attribution of anomalies through density-based methods, as we have shown in the submission. But is it a reasonable expectation?
> > > Invertible reparametrization does not fundamentally modify the problem (e.g., the density might change but the probability distribution does not fundamentally change through an invertible reparametrization: $P(X(\omega) \in A) = P(f(X(\omega)) \in f(A))$), but is merely looking at the same data through a different lens.
> > > For instance, transitioning from a cartesian coordinate system to a hyperspherical one shouldn’t change the way we consider the data. Yet, the resulting density from this reparametrization changes significantly from the original.
> > > For instance, the density of a standard Gaussian density is strictly positive everywhere in cartesian coordinates. However, when expressing this density in hyperspherical coordinates this density becomes zero over several hyperplanes. This comes from [the new infinitesimal volume in the hyperspherical coordinate system](https://en.wikipedia.org/wiki/N-sphere#Spherical_volume_and_area_elements).
> > >
> > > “Not as any contradiction, just to show again the example nicely shows: densities can, but do not necessarily correspond to what we intuitively would define as outliers.”
> > > We agree that there are cases where density-based methods fundamentally fail and others where it can succeed (as we wrote in the submission) depending on the representation and problem. Then why wouldn’t this criticism highlighting this critical dependence on problem and representation (stemming from this violation of invariance) be valid?
> > > Even if there is a philosophical disagreement on the nature of anomalies, the discussion emerging from this submission seems to have highlighted the discrepancy between what you call “model bias” of human perception and the default representation used, and how it results in density-based methods clashing with our intuitions of anomalies. As our conclusion emphasizes, the criticism focuses on the **agnostic** application of these density-based methods on the default representation.
> > > High-dimensional probabilities is in general an area where our intuition fails. Once again, the high-dimensional gaussian example illustrates that: the mode is a 0 while samples in that region are extremely rare if happening at all. This is why we cannot guarantee that CIFAR-10 and SVHN have disjoint support.
> > >
> > > 1/ The goal of our submission was to describe why this expectation of proper density scoring from a density model is in general unreasonable (and we agree that this problem goes beyond real-valued data). A representation where this expectation is met in a given problem is more of an exception than a rule. There is little reason to believe that density-based methods should result in a particular ranking of density that would separate SVHN and CIFAR-10 clearly.
> > >
> > > 2/ We think more realistic (i.e., complex) counterexamples can be constructed on a similar principle. The question is what would they bring more than the one-pixel example that we provided? What phenomenon would a more “realistic” example bring? For real-world problems, this issue from defining anomalies on the basis of density should be even more prevalent since anomalies can be present in the training set/distribution already.
> > >
> > > 3/ Yes, as we discussed in the conclusion, informing your anomaly detection method with some prior knowledge seems promising. It relies on the definition of anomalies as a *set* of outcomes whose probability (measure) is low. In this definition, being an anomaly is a property of the outcome and is invariant with respect to the representation (provided, chosen, or learned) (with invertible reparametrization).
> > >
> > > Please let us know if this addresses your concerns and, if so, we would appreciate it if you could reconsider your assessment. Regardless, we thank you for your time and effort in this discussion.

---

> > > > ### Comment · AnonReviewer4 · 2020-11-18
> > > > **Invertible reparametrization principle still seems unreasonable if one accepts only inlier samples as input in the first place**
> > > >
> > > > Thank you for your answer.
> > > >
> > > > I am still not sure I can agree with the invertible reparametrization principle as a desired and reasonable principle, and that it is analogous to the classification case. Allowing only access to samples of the reparameterized distribution destroys all information about the original inlier distribution except its support (meaning reparameterized samples that were originally outside original support will be outside reparameterized support), yes? Then to me, it seems a closer analogy in the classification case would be if you permute the labels for training and then expect the classifier to still work well on the original labels (as hinted by AnonReviewer1 as well), which it would not. So in other words, I am not sure I can agree with “Invertible reparametrization does not fundamentally modify the problem”, as only the information left is the support, and anomalies outside of support will continue to be detected (or is even that in contradiction to your point about the gaussian distribution in hyperspherical coordinates? - I did not completely follow there).
> > > >
> > > > One could also ask: Assuming an arbitrary invertible reparameterization and only access to samples after reparameterization, is there any method that can consistently detect anomalies? I am not sure, but seems more likely the answer is no? In other words, is there any method that can follow the invertible reparameterization principle? To me, given only access to inlier samples, it seems impossible? Except for anomalies outside support, but then the density-based method works as well. Regarding your definition as “a set of outcomes whose probability (measure) is low”. It seems to me either you are given this set a priori  - then you have anomaly labels, so a different setting - or you have a procedure for defining this set only based on inlier samples, then I suspect there should be no procedure that consistently yields the same sets across reparameterizations, or in other words there is no procedure that follows the invertible reparameterization principle.
> > > >
> > > > To me, the questions:
> > > > 1. When are density-based anomaly definitions consistent with human intuition?
> > > > 2. Are there other anomaly definitions which work consistently better or work better in real-world examples?
> > > > 3. Are there examples where one needs extra information (e.g., reference distribution for log-likelihood ratio or explicit anomaly labels) for sensible anomaly definition?
> > > > Are interesting, but different from what is currently discussed in the manuscript. This also concerns the pixel example, as it does not need the reparametrization principle as far as I see.
> > > >
> > > > I tend to disagree with SVHN vs CIFAR10 - imagine one correctly follows the CIFAR instructions to somehow get infinite data for CIFAR10 - there should never be any only-house-number-image inside, so support should be disjoint in my view.

---

> > > > > ### Author Response · Authors · 2020-11-19
> > > > > **Density estimation, a cornerstone of density-based approach, follows an invertible reparametrization principle**
> > > > >
> > > > > The depiction of invertible reparametrization as a destruction of information, including and especially about the original distribution, seems inaccurate. As written in our previous answer, the probability distribution does not change as its associated probability measure stays the same: $P_X(A) = P(X(\omega) \in A) = P(f(X(\omega)) \in f(A)) = P_{f(X)}(f(A))$. The reason why the density, defined as a Radon-Nikodym derivative, changes after reparametrization is that the base distribution changes: the Lebesgue measure in the original representation and the Lebesgue measure in the output representation are different. This change is reflected in the Jacobian determinant term of the change of variable formula. We need to accept this common ground about probability theory if we are to progress on this discussion.
> > > > > Moreover, if we add a continuity constraint on the principle (which we have no issue adding), several properties including openness and connectivity are conserved through these reparametrizations.
> > > > > We can also agree that density estimation, a cornerstone of density-based anomaly detection, follows an invertible reparametrization principle (albeit here, an equivariance principle). For example, if we transform the data distribution into a uniform (or any other distribution $q$), then, with *infinite data* and *infinite capacity*, a density model should match this uniform distribution (or the corresponding distribution $q$). Estimating density in a representation still allows one to recover the density in another representation (given invertible reparametrization). Once again, this is because invertible reparametrization conserves information about the distribution, but represents it differently.
> > > > >
> > > > > The example about hyperspherical coordinates is to provide a case where reparametrization more obviously describes the same vector than in the cartesian coordinates, yet modifies the probability density drastically. For example, it would create hyperplanes of density zero for a gaussian distribution (with dimension higher than 3). Thanks for pointing this out and allowing us to clarify.
> > > > >
> > > > > “One could also ask: Assuming an arbitrary invertible reparameterization and only access to samples after reparameterization, is there any method that can consistently detect anomalies? I am not sure, but seems more likely the answer is no? In other words, is there any method that can follow the invertible reparameterization principle? Except for anomalies outside support, but then the density-based method works as well.”
> > > > > That’s an excellent question we have provided some answers to in the conclusion. Likelihood ratio approaches satisfy this invariance principle: “a likelihood ratio method ([Ren et al., 2019](https://arxiv.org/abs/1906.02845); [Serra et al., 2020](https://openreview.net/forum?id=SyxIWpVYvr); [Schirrmeister et al., 2020](https://arxiv.org/abs/2006.10848)), which [...] is invariant to reparametrization”.
> > > > >
> > > > > The point of our submission (particularly in the conclusion) is that in order to have “density-based anomaly definitions consistent with human intuition” “one needs extra information (e.g., reference distribution for log-likelihood ratio or explicit anomaly labels) for sensible anomaly definition”. The reparametrization angle exposes the implicit but crucial assumption for a density-based approach to work (i.e. that the default representation used is the right one) and reframe this type of method as another anomaly detection approach using extra information. Once again, as stated in the conclusion of our submission, “the density scoring method has also been interpreted in [Bishop (1994)](http://citeseerx.ist.psu.edu/viewdoc/summary?doi=10.1.1.30.9127) as a likelihood ratio method” and we believe this interpretation is more transparent regarding the assumptions used in that approach (as opposed to an agnostic view of that method).
> > > > >
> > > > > “I tend to disagree with SVHN vs CIFAR10 - imagine one correctly follows the CIFAR instructions to somehow get infinite data for CIFAR10 - there should never be any only-house-number-image inside, so support should be disjoint in my view.”
> > > > > This is dependent on the noise model that you hypothesize on these datasets. For example, if one hypothesizes a small gaussian noise on the [camera sensors](https://en.wikipedia.org/wiki/Image_noise) for each dataset, suddenly the support of these distributions are not disjoint anymore.
> > > > > This is a good opportunity to state that the dataset comes from a pipeline we rarely fully control and that assuming that the resulting data will fit our hypothesis perfectly, i.e., assuming that the density model associated would yield scores that reflect the notion of anomaly, can be seen as unreasonable.

---

> > > > > > ### Comment · AnonReviewer4 · 2020-11-19
> > > > > > **The potential failures of maximum likelihood at anomaly detection are interesting but a different question from the invertible reparameterization to me**
> > > > > >
> > > > > > First, thank you for keeping a professional tone despite the long back-and-forth already, I  appreciate it :)
> > > > > >
> > > > > > I will restate the three issues as I see them now.
> > > > > >
> > > > > > 1. Reasonability of the invertible reparameterization principle
> > > > > >
> > > > > > First, I would claim if one uses density-based methods one assumes that the size of volumes in that space have meaning. This to me seems to remain the core issue, that me and other reviewers see it the same way: the usage of density-based methods implies the original representation has some meaning. Consider just as an example the typical way people dequantize discrete image data into continuous data by adding uniform noise around it such that the logarithm of integrating the continuous densities lower-bounds the logarithm of the discrete probabilities (https://arxiv.org/abs/1511.01844). One can argue about whether the original representation is ideal or not, but this is a different question from whether one expects the representation to be irrelevant when using density-based methods.
> > > > > >
> > > > > > Another point is that no method, not only density-based methods, that have only access to the invertibly reparameterized samples, without access to the invertible reparametrization, can meaningfully detect anomalies inside the support, since as you state the reparametrization can also result in the uniform distribution (this is the setting I meant , with no access to the invertible reparameterization, that destroys information). Note also that other tasks also become impossible, e.g., if you want to do clustering, if you just apply your clustering method on the uniform distribution, you cannot expect the same results as in the original space. If you instead allow access to the invertible reparametrization, one can compute the density in input space (as you also state in your reply as far as I understand)   using the log-determinant, which would also remove the problems with anomaly detection.
> > > > > >
> > > > > > I don’t think the fact that the probability that a point is in a certain set is the same as the probability that a transformed point is in the transformed set helps as you would need to know which sets you are interested in in the first place, which to me would be another example of an extra information.
> > > > > >
> > > > > > 2. Lack of practical insights from the fact that density-estimation methods do not obey the principle
> > > > > >
> > > > > > Further, I don’t see practical insights emerging from the invertible reparametrization principle issue, as the practical anomaly detection problems using generative models are the same for discrete cases, where the invertible reparametrization logic does not apply. Consider PixelCNN/PixelCNN++ which operate in discrete space and have the same behavior in Nalisnick et al., 2018 and Schirrmeister et al., 2020 on image datasets. Also you have the same issues in text, a naturally discrete data, as shown in Hendrycks et al. (2018).
> > > > > >
> > > > > > Also,in the typical image cases, turning the density method into an LLR method with the uniform distribution as the reference distribution would yield identical results despite now obeying the invertible reparameterization principle, again to me indicating this is maybe not a good angle to understand the practical issues from.

---

> > > > > > > ### Comment · AnonReviewer4 · 2020-11-19
> > > > > > > **[Continue]**
> > > > > > >
> > > > > > >
> > > > > > > 3. Manuscript is built around the failure of density-estimation methods to obey the principle
> > > > > > >
> > > > > > > As stated, the question of when purely training generative models by maximum likelihood is a good or bad way for anomaly detection is a good question to me (including as you state if one needs extra information, which I tend to actually agree with). However, the manuscript is not built around that, rather it is built around the invertible reparameterization principle with, in my view, all the issues described above.
> > > > > > >
> > > > > > > I think one can construct other interesting failure cases here similar to your pixel example, which again also work in the discrete case and therefore do not really relate well to the invertible reparametrization issue. For example, consider a case where one has a huge dataset of discrete pictures of all kinds  of birds and then every 10000th picture is black due to a camera malfunctioning - these black pictures would probably intuitively be seen as outliers, but will have a much higher probability (1/10000) then all actual bird pictures. But I don’t see a relation here to the whole invertible reparameterization thing, again as this is in the discrete setting. And I don’t think this applies to CIFAR10 vs SVHN, at least for CIFAR10 there was a human selection in the end - and also consider in discrete case, it would mean that whatever reason causes e.g., a house number to suddenly appear has to have higher probability then the typical CIFAR10 image - due to the very small probabilities of any concrete image it is hard to rule that out, but still seems unlikely to me.
> > > > > > >
> > > > > > > Also, if one wants to argue about reasonableness/unintuitive issues with maximum likelihood and anomalies I would expect there to be more discussion of what has already been written about it, e.g., see https://twitter.com/__ishaan/status/1300841526088658945 or https://twitter.com/sedielem/status/1325577058026741762 (“Apparently, a great deal of important semantic information lies in the ‘last few bits’ near the irreducible loss.”) https://benanne.github.io/2020/09/01/typicality.html
> > > > > > >
> > > > > > >
> > > > > > > Therefore, taken together, I don't want to endorse the manuscript as I don't see how the failure of density-based methods to follow the invertible reparameterization principle  substantially adds to understanding the issues with maximum-likelihood-based anomaly detection. Still I think thinking about the implicit assumptions of density-based methods or more generally maximum-likelihood-based methods (to cover the discrete case) and when they fail is certainly very interesting and a manuscript that discusses what has already been written about that and adds new insights would certainly be interesting.

---

> > > > > > > > ### Author Response · Authors · 2020-11-23
> > > > > > > > **Maximum likelihood pertains to inductive bias, which we eliminated in the "perfect density model" setting [1/3]**
> > > > > > > >
> > > > > > > > Thank you for continuing the discussion. We are glad that your point of view has evolved throughout the back-and-forth; now that you have pinpointed your concerns, we can answer and hopefully make things clear.
> > > > > > > >
> > > > > > > > 1/"the usage of density-based methods implies the original representation has some meaning. "
> > > > > > > > Indeed, you have mentioned that the bitmap was "not arbitrary, but a parametrization where we as humans can perceive the content"; we are questioning, however, "whether [...] the original [bitmap] representation is a good default representation for these [density-based] methods", since being meaningful for visualization does not imply that we should use these representations for anomaly detection. We have backed this dispute by providing a counter-example that showcases that the input representation does not necessarily have the right meaning for anomaly detection (the shades of black vs. white pixel example). This example demonstrates that density-based definitions of anomaly can differ from "intuitive" (i.e., salient for human perception) anomalies.
> > > > > > > >
> > > > > > > > We also challenged the claim (in your original point 3) that changing the representation meant changing the underlying outcome ("but I would argue they would also visually no longer be outliers") by providing the example of the change from Cartesian to hyperspherical coordinates, which, despite representing the same vectors, changes drastically the resulting density.
> > > > > > > >
> > > > > > > > Concerning the reparametrization principle: One way to define anomalies would be to attach the status to the root outcomes themselves irrespective of the representation. We show that density-based methods for anomaly detection are not compatible with that definition as they are tethered to a particular representation: A "density-based anomaly" is an "anomaly with respect to a choice of representation." Consistent with this, we pointed out that the default representation does not necessarily match an intuitive (e.g., visual) definition of anomalies; we even provided a meaningful counter-example where intuitive anomalies are not density-based anomalies (the shades of black vs. white pixel example).
> > > > > > > >
> > > > > > > > More importantly, we show that it is not guaranteed--even with infinite data and capacity--that a density-based method will result in a reasonable (e.g., "intuitive", "visual") separation of inliers and outliers, without any prior knowledge or assumption. Coming back to the classification analogy, standard results show that in the limit of infinite data and capacity, several widely used machine learning algorithms converge to the right solution. The fact that density-based methods for anomaly detection do not respect this bare minimum for a machine learning algorithm should be concerning for a practitioner. The field considers impossibility results, [including in clustering](https://www.cs.cornell.edu/home/kleinber/nips15.pdf), important.
> > > > > > > >
> > > > > > > > The impossibility of *guaranteeing* intuitive anomalies even with infinite data and capacity is a *representation* problem since intuitive anomalies can be recovered by choosing the right representation; however, this requires prior knowledge.
> > > > > > > >
> > > > > > > > "I don't think the fact that the probability that a point is in a certain set is the same as the probability that a transformed point is in the transformed set helps as you would need to know which sets you are interested in in the first place, which to me would be another example of an extra information."
> > > > > > > > Once again, we pointed this out to show that the probability measure associated with a distribution does not change through reparametrization to challenge your statement that an invertible reparametrization destroys information about the original distribution.
> > > > > > > >
> > > > > > > > And yes, we agree that one would need additional information (what we refer to as "prior knowledge" or "assumptions") to discover the set of anomalies; this is, in fact, a point that we've written in our conclusion (as reviewer 2 as observed). But, once again, we do not frame the use of prior knowledge or assumptions as a "bad thing", but instead an assumption that should be made explicit and that should inform our design of (at least) density-based methods for anomaly detection. This brings us to the next point.

---

> > > > > > > > > ### Author Response · Authors · 2020-11-23
> > > > > > > > > **Maximum likelihood pertains to inductive bias, which we eliminated in the "perfect density model" setting [2/3]**
> > > > > > > > >
> > > > > > > > >
> > > > > > > > > 2/ We show in our paper that density-based approaches that seemingly do not rely on prior knowledge and that people tend to use off the shelf, such as density scoring or a typicality test, cannot work on any arbitrary representation (input representation, embedding, …). The aim of our paper is, therefore, to highlight this hidden assumption (which has itself little justification). For example:
> > > > > > > > >
> > > > > > > > > "the density method [can be seen as a] LLR method with the uniform distribution as the reference distribution would yield identical results despite now obeying the invertible reparameterization principle, again to me indicating this is maybe not a good angle to understand the practical issues from."
> > > > > > > > > As mentioned in the conclusion of our submission, "density scoring method has also been interpreted in Bishop (1994) as a likelihood ratio method (Ren et al., 2019; Serrà et al., 2020; Schirrmeister et al., 2020), which, on the one hand, relies heavily on the definition of an arbitrary reference density as a denominator of this ratio but, on the other hand, is invariant to reparametrization." We also added in our last reply "we believe this interpretation [of density scoring] is more transparent regarding the assumptions used in that approach (as opposed to an agnostic view of that method)." One question that comes naturally from exposing density scoring as a likelihood ratio method with respect to the uniform distribution is: is the uniform distribution a reasonable denominator to use? Are there better background distributions to use in this case? Ren et al. (2018) and Serra et al. (2019) seem to say "yes."
> > > > > > > > >
> > > > > > > > > "Consider PixelCNN/PixelCNN++ which operate in discrete space and have the same behavior in Nalisnick et al., 2018 and Schirrmeister et al., 2020 on image datasets."
> > > > > > > > > The scope of our paper is continuous data, which we think is broad enough to have a practical impact. The same discrepancy as the one we highlighted in the case of continuous data in our submission might happen with discrete data but for different reasons, as the reparametrization does not apply here indeed.
> > > > > > > > >
> > > > > > > > > However, in the case of images, the bitmap representation is a discretization of a continuous domain. Therefore we believe our analysis holds some validity in this case. For example, the way we build our one-pixel counter-example has been to reweight out the bimodal black and white distribution you mentioned and spread out the mode corresponding to the color black, very similarly to a reparametrizing approach.

---

> > > > > > > > > > ### Author Response · Authors · 2020-11-23
> > > > > > > > > > **Maximum likelihood pertains to inductive bias, which we eliminated in the "perfect density model" setting [3/3]**
> > > > > > > > > >
> > > > > > > > > > 3/"purely training generative models by maximum likelihood is a good or bad way for anomaly detection is a good question to me"
> > > > > > > > > > This notion pertains to the issues of "estimation, approximation, or optimization errors" (that we mentioned in the submission). We have removed them from the equation by considering **perfect density models** (i.e., $p^{(\theta)}_X = p^*_X$). We explicitly pointed out first in our submission: "This [infinite data and capacity] setting is appealing as it gives space for theoretical results without worrying about the underfitting or overfitting issues mentioned by Hendrycks et al. (2018); Morningstar et al. (2020); Kirichenko et al. (2020); Zhang et al. (2020) "and in our first reply "At least Kirichenko et al. (2020) mention by name "inductive bias" which suggests to us that, in this narrative, this failure of density models for anomaly detection would be solved if a perfect density model was provided. We show in general (beyond the image case) that it's not true".
> > > > > > > > > >
> > > > > > > > > > [In the first tweet you mention](https://twitter.com/__ishaan/status/1300841526088658945), Ishaan cites this [paper](https://web.mit.edu/cocosci/Papers/coincidences5.pdf) which discusses likelihood ratio and bayesian methods (which we mention in the conclusion).
> > > > > > > > > >
> > > > > > > > > > This quote ["Apparently, a great deal of important semantic information lies in the 'last few bits' near the irreducible loss."](https://twitter.com/sedielem/status/1325577058026741762) is likely to pertain to inductive bias as the paper cited is titled "Scaling Laws for **Autoregressive Generative Modeling**" and standard autoregressive modelling is biased toward learning local information first. This contrasts with some fully-connected models, like fully-connected variational autoencoders or DRAW, whose samples were described as blurry because they seem to prioritize global information that one would consider more "semantic."
> > > > > > > > > >
> > > > > > > > > > Moreover, [Sander's blogpost](https://benanne.github.io/2020/09/01/typicality.html) (which we cite) mention the typicality method, which is also a density-based method that we also study in our paper. This blogpost also mentions the idea "measuring likelihood in more abstract representation spaces" to succeed, and we describe more accurately the mechanism through which one can achieve this. While this mechanism of change of variable has appeared in [Nalisnick, et al. (2018)](https://arxiv.org/abs/1810.09136), it was mainly used as a decomposition of the likelihood term in flow models, not as an approach to building general counter-examples.
> > > > > > > > > >
> > > > > > > > > > "Still I think thinking about the implicit assumptions of density-based methods [...] and when they fail is certainly very interesting, and a manuscript that discusses what has already been written about that and adds new insights would certainly be interesting."
> > > > > > > > > > Thank you. We indeed discuss what has already been written about "density-based methods [...] and when they fail" and add "new insights" about how several of these hypotheses are insufficient, since even solving the inductive bias problem, being in a low-dimensional setting, or using the typicality method as in Nalisnick, et al. (2019) won't address this new issue of representation.
> > > > > > > > > >
> > > > > > > > > > "density-based methods or more generally maximum-likelihood-based methods (to cover the discrete case)"
> > > > > > > > > > Maximum likelihood training of probabilistic models is not the discrete equivalent to density-based methods to anomaly detection: the former pertains to the way a model does not fit exactly the target distribution. A discrete equivalent would be probability-based methods.
> > > > > > > > > >
> > > > > > > > > > Please let us know if these clarifications address your concerns.

---

> ### Author Response · Authors · 2020-11-16
> **Addressing your points (2/2)**
>
>
> For the other point, [Schölkopf et al. (2001)](https://citeseerx.ist.psu.edu/viewdoc/download?doi=10.1.1.60.9423&rep=rep1&type=pdf) is an example of a setting where “one would usually pick the set of regular points X_in such that this set contains the majority (but not all) of the mass (e.g.,95%) of this distribution”. We will cite it again next to the statement.
> ““Without any constraints on the space considered, individual densities can be arbitrarily manipulated, which reveals how little these quantities mean in general.” -> I don’t agree this shows how little densities mean in general. I think it shows how little they mean once you allow arbitrary invertible reparametrizations, and that maybe allowing this is not a good idea ;)”
> There are a couple of words missing there and propose the following edit: “how little these quantities [say about the underlying outcome] in general”. Thanks for bringing this to our attention, we do not want to ignore, for example, the link between negative log-likelihood and Kullback-Leibler divergence. While this is true that one can create an adversarial situation knowing the true density and the regularity scoring with access to arbitrary invertible reparametrization, we should be aware that one can already be in one of those adversarial situations unknowingly. We invite you to read our argument [**here**](https://openreview.net/forum?id=MkrAyYVmt7b&noteId=HxfVX36hwJZ).
>
> “if they are not defined by density-level at original parametrization, then the question is what are they defined by anyways, and you have a whole new question in my view.”
> We are glad that our paper was able to convey this question clearly enough and invite you to read [Schirrmeister et al. (2020)](https://arxiv.org/abs/2006.10848), [Kirichenko et al. (2020)](https://arxiv.org/abs/2006.08545), [Winkens et al. (2020)](https://arxiv.org/abs/2007.05566), and [Lee et al. (2018)](https://arxiv.org/abs/1807.03888) for methods using densities in another representation. Once again, while these examples are more empirical, their interpretations suffer from the ambiguity stemming from the unsolvable problem of learning the perfect density models with a finite amount of data. Our paper focuses on a more theoretical ground where the problem of density estimation (including inductive biases, which have been extensively discussed in previous papers) can be safely dismissed and identify a more fundamental problem with these density-based methods.

---

### Official Review · AnonReviewer1 · 2020-10-26
**While the paper has interesting discussions and constructions of the worst case scenarios for confusing density-based anomaly detection methods, I don’t think that the paper uses them to support a useful claim.**

**Rating:** 4
**Confidence:** 4

**Review:**

*Quality*
I have a major concern regarding the principle stated on the page 4 that the whole paper discussion is based on.
First, I totally disagree that anomaly detection methods should be invariant under any invertible reparametrizations. In practice, quite often the very definition of an anomaly detection problem is tighted to a concrete data representation.
In addition, anomaly detection can be thought of as a binary classification and the stated principle amounts to saying that a binary classifier should provide the same results for any (invertible) data re-parametrization, e.g. using various feature extractors. This is clearly wrong thing to ask from a classifier and instead people focus on finding useful feature extractors. Hence, in the case of anomaly detection, it comes at no surprise that one can re-parametrize the data so that the anomaly detection algorithm (density or not density based) is totally confused.

While the paper has interesting discussions and constructions of the worst case scenarios for confusing density-based anomaly detection methods, I don’t think that the paper uses them to support a useful claim.

*Clarity*
The paper is well-written in general.

*Originality*
To my best knowledge, the theoretical study of the role of data re-parametrization for density-based anomaly detection is new.

*Significance*
I don’t think that the results of the paper are significant in their current form and might cause a confusion with its statements.

I would encourage the authors to clarify their view on the relation between anomaly detection and the underlying data representation and clarify/justify the principle from the page 4.


Pros
* The paper presents a collection of interesting transformations between densities

Cons
* The main message of the paper is based on unrealistic principle

---

> ### Author Response · Authors · 2020-11-16
> **About the invariance principle in binary classification**
>
> Thank you for expressing your concerns with this submission.
> We disagree on the premise that the statement that “a binary classifier should provide the same results for any (invertible) data re-parametrization [...] is clearly wrong”. We reiterate the argument we wrote [**here**](https://openreview.net/forum?id=MkrAyYVmt7b&noteId=HxfVX36hwJZ).
> Since the attribution of inlier and outlier labels is unknown a priori (although we can impose mild regularity conditions), we want a reasonable anomaly detection approach to learn, in every possible case, the right solution given enough evidence and capacity. This is one of the reasons why it is important to have a universal approximation theorem for the set of considered functions. Learning classifiers with deep learning is an example of that: otherwise attempts at solving this task would be hopeless.
> In classification, the invariance principle with infinite data and capacity should apply. Otherwise, learning classifiers on encrypted data (as in Crypto-Nets; Xie, et al., 2014) would be unreasonable, but the encrypted data of the image of a horse still represents a horse. As we describe in section 3, we consider outlier detection to be a binary classification problem (albeit without supervision) and we assume that the status of outlier / inlier (i.e., the regularity) is a latent label attributed to each *outcome*, regardless of representation. Therefore we believe the principle of invariance from section 3 should apply to anomaly detection.
> We do agree with you, that people do “focus on finding useful feature extractors”, but our understanding of these approaches is that it relates to capacity and learnability rather than consistency.

---

> > ### Comment · AnonReviewer1 · 2020-11-17
> > **Anomaly detection is inherently coupled with representation**
> >
> > Thanks for you reply. I agree with a comment of AnonReviewer4 below that you should be clearer on what anomalies are for you. First, typically anomalies are observations that do not follow the regular data distribution and thus fall in the low probability regions. Density estimation is just an approach to find these regions. This is inherently connected to the data representation, so invariance principle does not seem to be useful as the problem setup itself is not invariant to the representation.
> > Second, if you define anomalies as low probability observations in the original sample space (omega), then again, this definition is coupled with the choice of the representation, it is just an identity mapping in this case.
> > Your example of classification on encrypted data does not really work for anomaly detection as in classification the data representation does not change the class label.

---

> > > ### Author Response · Authors · 2020-11-19
> > > ***Density-based* anomaly detection is inherently coupled with representation**
> > >
> > > Thank you for continuing the discussion.
> > > An important distinction is the definition of probability of subsets vs density of points. The probability (measure) of a subset is the integral of the density of the points it contains. A point have a density and the probability (measure) associated with its singleton is in general (if its density is finite) 0.
> > >
> > > We indeed define anomalies as elements of “low probability regions”, a subset A with low probability **measure**. Using the density scores is certainly *one* way to define such subset A in a way that is inherently tied with the representation, but certainly not the only one (see [Schölkopf et al. (2001)](https://citeseerx.ist.psu.edu/viewdoc/download?doi=10.1.1.60.9423&rep=rep1&type=pdf) and our Figure 1).
> > > Regarding density estimation, it follows a principle related to invertible reparametrization. For example, if we transform the data distribution into a uniform (or any other distribution $q$), then, with *infinite data* and *infinite capacity*, a density model should match this uniform distribution (or the corresponding distribution $q$). Approaches like normalizing flows rely on this fact.
> > >
> > > “in classification the data representation does not change the class label.”
> > > Indeed, and if one defines anomaly as belonging to this subset A of low probability measure, then neither should the data representation (with invertible reparametrization) in anomaly detection: $x \in A \Rightarrow f(x) \in f(A)$ and $P_X(A) = P_{f(X)}(f(A))$.
> > >
> > > The definition of the set of anomalies A depends on the problem and an intuitive definition of these anomalies does not necessarily match density-based definitions with the default representation.
> > >
> > > Please let us if you agree with the points developed above. Otherwise, feel free to point out any gap in our reasoning. Thank you!

---

### Official Review · AnonReviewer2 · 2020-10-29
**Invariance is not a required condition for density estimation.**

**Rating:** 4
**Confidence:** 4

**Review:**

This paper presents a negative result that anomaly detection by density estimation is flawed.

After highlighting several problems with anomaly detection, the paper focuses on a seemingly plausible invariance argument, that the lack of invariance of a distribution's random variable to change of variable - reparameterization - makes it impossible to define which events fall in dense or rare areas of the distribution. Specifically it proposes the principle that an outcome represented by a random variable X or it's transformation fX should not result in change of it's classification as outlier versus inlier.  As the paper demonstrates, it is not difficult to devise a counter example of a transform f that can flip an outcome between inlier and outlier with weak conditions on any distribution of X.

 The paper identifies inliers by introducing "typicality" as a measure. Although interesting, it contributes little to the paper's argument and unlike the level set definition of inliers, typicality does give not a constructive tool for anomaly detection.

 However is any practical sense, the invariance argument is specious. Consider this example, of the distribution of sunlight over the earth. If one is interested in the effects of sunlight on population health (e.g. cancer rates) one looks at the distribution of intensity over individuals. Alternately if the question is the incidence over area (solar power potential) then its the random variable of land area that is relevant.  There is an invertable deterministic function between land area and population random variables given by the population per area . Clearly a place (think outcome) with low cancer incidence (an outlier) because of sparse population may be a good place for solar power. Hence the parameterization of random variable for anomaly detection is intrinsic to the problem rather than serving as a condition for invariance.

 In a related area, invariance as applied to prior distributions has been studied extensively, as found in E. T. Jaynes, "Probability Theory" (Cambridge, 2003), ch 12, to show when invariance is applicable.

I have no issue with the paper's rigor and derivations, rather the issue is with the unfounded presumption that invariance to reparameterization inhibits the use of density based anomaly detection. As the paper states in the final discussion, "defining anomalies might be impossible without prior knowledge"-- this may not be a bad thing, and the same material used to argue for a knowledge-based approach to anomaly detection that includes specifying a random variable appropriate for the domain would be a valuable, acceptable paper.

---

> ### Author Response · Authors · 2020-11-16
> **Clarifying a couple of points**
>
> Thank you for sharing your perspectives.
> We want to clarify a couple of points:
> - we agree that “invariance is not a required condition for density estimation”, this invariance principle would apply to anomaly detection methods (here density-based anomaly detection). We do not criticize the concept of density estimation but its use in the context of anomaly detection;
> - we do not want to claim that we are “introducing "typicality" as a measure” for anomalies, [Nalisnick et al. (2019)](https://arxiv.org/abs/1906.02994) does introduce it as “a constructive tool for anomaly detection”.
>
> Please let us know if there is any part of the submission that created any confusion.
> The choice of parametrization is crucial to ensure that density-based approaches succeed at detection anomalies, as we discussed in section 4. However, we wish to challenge the assumption that the original input representation provided in an anomaly detection task is necessarily a good fit in this context. We develop this argument further [**here**](https://openreview.net/forum?id=MkrAyYVmt7b&noteId=HxfVX36hwJZ). In an ideal case, anomaly detection methods should be able to circumvent issues related to representation with enough data and capacity and converge to the right solution, but density-based methods (as is) do not seem to necessarily be able to. We agree that “defining anomalies might be impossible without prior knowledge” may not be a bad thing, but we wish to emphasize that remark nonetheless as several practitioners are set to use these density-based methods off-the-shelf without considering much of the issues discussed in the submission.

---

### Official Review · AnonReviewer3 · 2020-10-29
**The paper offers a critique of the guarantees density models can provide when used for anomaly detection. While I support a push for formal definitions and requirements I find the critique itself off the mark.**

**Rating:** 3
**Confidence:** 4

**Review:**

Detecting anomalies is a notoriously ill-defined problem. The notion of anomaly is not a rigorous concept and different algorithms produce different results. The paper critiques a broad set of methods which involve likelihood (or density) estimations. It's main idea revolves around the 'Principle' set on Page 4. The principle claims that when data capacity and computational constraints are removed, an AD algorithm should be invariant to 'reparametrization' of the input. Roughly speaking, that means the algorithm should be invariant to arbitrary 'name changing' of the input - the result should not change if each data item x is replaced by f(x) if f is invertible.
The paper then shows that density models do not satisfy this principle even when they are 'perfect'.

My main critique of the paper is that this principal constitutes a completely unreasonable requirement of any AD algorithm, to the point where it is meaningless. It is trivial to observe (as the authors do) that any continuous distribution could be transformed to a uniform distribution with the correct f. Even if the domain is discrete we can make the input uniform: As an example think of f being a Pseudo-Random function, like say f(x) is a digital signature of x. If we believe cryptography then no efficient algorithm would be able to say anything useful.

To sum up - I don't think the 'principle' is a useful prism by which to measure models and definitions, and therefore I don't find the contribution of the paper sufficient for publication.

As a side remark, this principle may be useful if further constraints are put on f, for instance, if we may want the AD algorithm to be oblivious to unit change in the data, which translates to f which multiplies dimension be a constant.

---

> ### Author Response · Authors · 2020-11-16
> **Answer to AnonReviewer3**
>
> Thank you for expressing your concerns with this submission.
> We are glad that we agree on the extent of the problems that these reparametrizations entail for anomaly detection. While we agree that constraining the array of reparametrizations can limit the ability to create adversarial situations for density-based anomaly detection, there is no way to know whether the original input representation is not one of these adversarial situations. We develop this argument further [**here**](https://openreview.net/forum?id=MkrAyYVmt7b&noteId=HxfVX36hwJZ), please let us know if this argument is unclear or there is any oversight in our reasoning.

---

> > ### Comment · AnonReviewer3 · 2020-11-18
> > **The 'invariance principal' is misguided**
> >
> > Thank you for engaging and for your thoughtful comments. I have read carefully the responses, but I'm still convinced the 'invariance principal' as stated is such a strong requirement it renders the entire anomaly detection task meaningless. Most data sets are sparse, in that no point appears more than once, in this case reparameterizaton allows you to map any point to any point making notions of classification or anomaly meaningless. In other words,  anomaly, like any other classification, is not a function of the point itself but of the region in space within which it lies. Put differently, an image is labeled as 'dog' because it resembles other dogs not because that exact image was once labeled before.
> > I'm afraid I'm not changing the recommendation to reject.

---

> > > ### Author Response · Authors · 2020-11-19
> > > **We will re-emphasize the continuity of the example reparametrizations**
> > >
> > > Thank you for continuing the discussion. Allow us to contextualize your concerns in a more formal setting.
> > >
> > > "Most data sets are sparse"
> > > We want to re-emphasize that the principle that we use is in the **infinite data** and **infinite capacity** setting, where one should hope to be able to obtain the exact solution. The presence of *universal approximation theorems* and *no bad minima results* for deep learning are motivated by a desire to satisfy similar principles.
> > >
> > > "In other words, anomaly, like any other classification, is not a function of the point itself but of the region in space within which it lies."
> > > In our paper the reparametrizations that we provide as examples are continuous (homeomorphisms), which means we can modify the principle to add this additional continuity constraint. With that constraint, the region within which a point lies will stay the same: a "dog" that was path-connected to another "dog" within the "dog" set before the homeomorphism will stay path-connected within the set of "dog"s after the homeomorphism, and a "dog" that has "dog"s within $\epsilon $distance for every $\epsilon > 0$ before homeomorphism will still have "dog"s within $\epsilon$ distance for every $\epsilon > 0$ after homeomorphism.
> > >
> > > We think this would satisfy the concerns you brought forward. Is there an alternative formalism that you were referring to?

---

### Author Response · Authors · 2020-11-13
**(R1, R2, R3, R4) Justification of the invariance principle.**

We want to thank all the reviewers for their time, helpful feedback, and advice. We appreciate the reviewers challenging our statements and helping us make our paper stronger.
All reviewers asked for explanations on the invariance principle, we hope to address your concerns in the following rebuttal. Otherwise, let’s discuss these further.

###  Why is the ability to express a large set of labelling desirable (in an infinite data and capacity setting)?
Since the attribution of inlier and outlier labels is unknown a priori, we want a reasonable anomaly detection approach to learn, in every possible case (under mild regularity conditions), the right solution given enough evidence and capacity. This is one of the reasons why it is important to have a universal approximation theorem for the set of considered functions. Learning classifiers with deep learning is an example of that: otherwise attempts at solving this task would be hopeless.
### Why is invariance desirable?
In classification, the invariance principle with infinite data and capacity should apply. Otherwise, learning classifiers on encrypted data (as in [Crypto-Nets; Xie, et al., 2014](https://arxiv.org/abs/1412.6181)) would be unreasonable, but the encrypted data of a horse still represents a horse, you just need to decrypt this data to visualize it again. As we describe in section 3, we consider outlier detection to be a binary classification problem (albeit without supervision) and we assume that the status of outlier / inlier (i.e., the regularity) is a latent label attributed to each *outcome*, regardless of representation (with the same information). Therefore we believe the principle of invariance from section 3 should apply to anomaly detection.
### Why can density-based approaches applied to the original input representation be incorrect in the ideal case (infinite data and capacity)?
As all of the reviewers seem to disagree with that principle, we suspect (but please correct us if we misunderstood) that their alternative definition would be that regularity is a property best defined by $p(x)$ in the original representation, e.g., the bitmap representation for images. We propose several points challenging this definition:

1/ [Schirrmeister et al. (2020)](https://arxiv.org/abs/2006.10848) and [Kirichenko et al. (2020)](https://arxiv.org/abs/2006.08545) have shown an improvement of the performance of density scoring when used on alternative representations, which shows that the original input representation might be suboptimal, although this could be attributed to an inductive bias or a learnability issue of their model.

2/ Let’s take a one pixel image for example to not only fit the bitmap representation narrative but also to eliminate the hypothesis of texture bias inside convolutional models as a cause of failure. Let’s consider the following mixture distribution: $U([255, 256]^3) $(white) with mixture probability 0.01 and $U([0, 10]^3)$ (shades of black) with mixture probability 0.99. If we put every sample side to side in one image, we obtain the following image (https://imgur.com/a/bNOvBJl), where one would consider the white colored pixels to be outliers. However, these white values have a density of 0.01 each while darker values have a density of 0.00099 (<0.01) each. The same goes for probabilities in a {0, …, 255} discretization of this counter-example. In this bitmap representation, the density values go against the intuition of associating density values with anomaly detection, despite knowing the underlying density model perfectly (therefore no inductive bias involved).

3/ While bitmap is indeed a representation that allows us to visualize pictures on a screen, it does not have to mean that it should necessarily be taken as reference. For instance, a 3D scene is usually visualized through 2D images but it does not mean that a bitmap representation of this 2D image is the right one for a 3D scene that contains much more information.

If bitmap is not necessarily the ideal representation for images and scenes for density-based anomaly detection methods, we can also wonder whether, in the general case, the original representation is a good default representation for these methods. We do not mean that "defining anomalies might be impossible without prior knowledge" is “a bad thing”, but want practitioners to be aware of this caveat before using these density-based methods off-the-shelf: while often understated, the assumption that the original input representation will yield correct density values for anomaly detection in an ideal (infinite data and capacity) case is a strong one. Prior knowledge can take different forms like distribution of inliers/distribution of outliers (e.g., in likelihood ratio approaches) or prior distribution on density models and parametrizations (e.g., in a bayesian setting).

If any of these points is unclear or unsatisfactory, please feel free to continue the discussion here.

---

> ### Comment · Area_Chair1 · 2020-11-18
> **Authors: Thank you for response / Reviewers: Please update**
>
> Thank you, authors, for your responses.  And thanks R1 and R4 for your follow-up discussion.
>
> R2 and R3, please read the responses and update your reviews by stating that your concerns have been addressed or by providing further rebuttal.

---

### Public Comment · ~Jianwen_Xie1 · 2020-11-14
**missing related works about deep density model estimation**

Dear Authors,

We found that the current paper missed some important references about pioneering works that are related to deep density model or energy-based models parameterized with deep net energy.

The first paper that proposes to train an energy-based model parameterized by modern deep neural network and learned it by Langevin based MLE is in (Xie. ICML 2016) [1]. The model is called generative ConvNet, because it can be derived from the discriminative ConvNet. This is also the first paper to formulate modern ConvNet-parametrized EBM as exponential tilting of a reference distribution, and connect it to discriminative ConvNet classifier. That is, EBM is a generative version of a discriminator. (Xie. ICML 2016) [1] originally studied such an EBM model on image generation theoretically and practically in 2016.

(Xie. CVPR 2017) [2] (Xie. PAMI 2019) [3] proposed to use Spatial-Temporal ConvNet as the energy function in EBMs for video generation.

(Xie. CVPR 2018) [4] also proposed to use volumetric 3D ConvNet as the energy function for 3D shape pattern generation. It is called 3D descriptor Net.

Also, the Generative Cooperative Nets (CoopNets) (Xie. PAMI 2018)[5] and (Xie. AAAI 2018) [6], which jointly trains an EBM and a generator network by MCMC teaching. This is also about estimation of density models.

Those are the more original and earlier papers for deep EBMs with ConvNet as energy function than what you have cited, e.g., [7](Yilun Du and Igor Mordatch, 2019).

References:

[1] A Theory of Generative ConvNet. Jianwen Xie *, Yang Lu *, Song-Chun Zhu, Ying Nian Wu (ICML 2016)

[2] Synthesizing Dynamic Pattern by Spatial-Temporal Generative ConvNet Jianwen Xie, Song-Chun Zhu, Ying Nian Wu (CVPR 2017)

[3] Learning Energy-based Spatial-Temporal Generative ConvNet for Dynamic Patterns Jianwen Xie, Song-Chun Zhu, Ying Nian Wu IEEE Transactions on Pattern Analysis and Machine Intelligence (TPAMI) 2019

[4] Learning Descriptor Networks for 3D Shape Synthesis and Analysis Jianwen Xie *, Zilong Zheng *, Ruiqi Gao, Wenguan Wang, Song-Chun Zhu, Ying Nian Wu (CVPR) 2018

[5] Cooperative Training of Descriptor and Generator Networks. Jianwen Xie, Yang Lu, Ruiqi Gao, Song-Chun Zhu, Ying Nian Wu. IEEE Transactions on Pattern Analysis and Machine Intelligence (TPAMI) 2018

[6] Cooperative Learning of Energy-Based Model and Latent Variable Model via MCMC Teaching. Jianwen Xie, Yang Lu, Ruiqi Gao, Ying Nian Wu. AAAI 2018.

[7] Yilun Du and Igor Mordatch. Implicit generation and modeling with energy based models. In Advances in Neural Information Processing Systems, pages 3603–3613, 2019

Thank you!

---

> ### Author Response · Authors · 2020-11-14
> **Do you want us to cite your work? We'll be happy to.**
>
> Dear Jianwen Xie,
>
> Congratulations on spearheading this pioneering work on deep density models and energy-based models and thank you for taking the time to look at our submission!
> Given the context of our work (anomaly detection), I hope you can understand that we are citing [Du and Mordatch (2019)](https://arxiv.org/abs/1903.08689) criticizing their evaluation procedure of their probabilistic model using an out-of-distribution detection metric. We have taken a quick glance at the papers that you have listed and have found no similar evaluation procedure. If there is and we have missed it, we will be happy to add these papers to comment on the problematic nature of their evaluation methods.
>
> Thank you!

---

> > ### Comment · Area_Chair1 · 2020-11-18
> > **EBM Work Not Related**
> >
> > I agree with the authors that the above listed works on energy based models (EMBs) are indeed not relevant (unless there's discussion of anomaly / out-of-distribution detection).  The paper's citations to the EBM literature are related to out-of-distribution detection, not energy-based modeling.

---

### Decision · Program_Chairs · 2021-01-07
**Final Decision**

**Decision:**

Reject

**Comment:**

Firstly, thank you authors for your thought-provoking submission and discussion.  The key point of disagreement clearly is the fundamental assumption that "the result of an anomaly detection method should be invariant to any continuous invertible reparametrization f."  All reviewers found this assumption to be too strong, leading all four to recommend rejection.

I also recommend rejection at this time.  To me, it seems reasonable and practical to assume that anomalies are defined based on distance (in a fixed feature space).  So if we are allowed to deform the space, clearly this definition breaks down and the concept of an anomaly becomes empty.  Perhaps I am wrong about this, but nevertheless, the paper could do a much better job of convincing the reader that its fundamental reparametrization assumption is appropriate and of consequence in practice.